# Chemical Composition and Biological Activities of Metabolites from the Marine Fungi *Penicillium* sp. Isolated from Sediments of Co To Island, Vietnam

**DOI:** 10.3390/molecules24213830

**Published:** 2019-10-24

**Authors:** Hong Minh Thi Le, Quynh Thi Do, Mai Huong Thi Doan, Quyen Thi Vu, Mai Anh Nguyen, Thu Huyen Thi Vu, Hai Dang Nguyen, Nguyen Thi Thuy Duong, Manh Hung Tran, Van Minh Chau, Van Cuong Pham

**Affiliations:** 1Institute of Marine Biochemistry, Vietnam Academy of Science and Technology, 18 Hoang Quoc Viet, CauGiay, Hanoi 100803, Vietnam; quynhdt1992@gmail.com (Q.T.D.); huongdm@imbc.vast.vn (M.H.T.D.); svuquyen@yahoo.com (Q.T.V.); maianhhsb@gmail.com (M.A.N.); huyenvuibt@gmail.com (T.H.T.V.); minhcv@vast.vn (V.M.C.); phamvc@imbc.vast.vn (V.C.P.); 2Graduate University of Science and Technology, Vietnam Academy of Science and Technology, 18 Hoang Quoc Viet, CauGiay, Hanoi 100803, Vietnam; 3University of Science and Technology of Hanoi (U.S.T.H.), Vietnam Academy of Science and Technology, 18 Hoang Quoc Viet, CauGiay, Hanoi 100803, Vietnam; nguyenhd@imbc.vast.vn; 4Biomedical Science Department, Institute for Research & Executive Education (V.N.U.K.), The University of Danang, 158A Le Loi, Hai Chau District, Danang City 551000, Vietnam; duong.nguyen@vnuk.edu.vn

**Keywords:** marine fungi *Penicillium* sp., 3-acetyl-4-hydroxycinnoline, 4-hydroxybenzandehit, antimicrobiology, α-glucosidase

## Abstract

Marine microorganisms are an invaluable source of novel active secondary metabolites possessing various biological activities. In this study, the extraction and isolation of the marine sediment *Penicillium* species collected in Vietnam yielded ten secondary metabolites, including sporogen AO-1 (**1**), 3-indolecarbaldehyde (**2**), 2-[(5-methyl-1,4-dioxan-2-yl)methoxy]ethanol (**3**), 2-[(2*R*-hydroxypropanoyl)amino]benzamide (**4**), 4-hydroxybenzandehyde (**5**), chrysogine (**6**), 3-acetyl-4-hydroxycinnoline (**7**), acid 1H-indole-3-acetic (**8**), cyclo (Tyr-Trp) (**9**), and 2’,3’-dihydrosorbicillin (**10**). Their structures were identified by the analysis of 1D and 2D NMR data. Among the isolated compounds, 2-[(5-methyl-1,4-dioxan-2-yl)methoxy]ethanol (**3**) showed a strong inhibitory effect against *Enterococcus faecalis* with a minimum inhibitory concentration value of 32 µg/mL. Both 2-[(2*R*-hydroxypropanoyl)amino]benzamide (**4**) and 4-hydroxybenzandehyde (**5**) selectively inhibited *E. coli* with minimum inhibitory concentration values of 16 and 8 µg/mL, respectively. 2’,3’-Dihydrosorbicillin (**10**) potentially inhibited α-glucosidase activity at a concentration of 2.0 mM (66.31%).

## 1. Introduction

Drug-resistant bacteria have been an emerging global problem in the last few decades and are one of the most serious issues affecting public health. In the treatment of infections, there is a sharp increase in the number of pathogens with multiple drug-resistant agents [1]. Among them, biofilm-synthetic small molecules are microbes that survive in hostile environments and raise the resistance by a thousand-fold. Biofilms potential to cause disease often occurs in Gram-negative pathogens such as *Streptococcus pneumoniae*, *Staphylococcus aureus*, and *Enterococcus faecium*, as well as Gram-negative pathogens such as *Escherichia coli* and *Pseudomonas aeruginosa* [2]. Of particular note is some bacteria are naturally resistant to a certain antibiotic, but otherwise, antibiotic resistance is the result of the long-term use of antibiotics to treat infections [3]. Recent research revealed that antimicrobial chemotherapy could prevent the pathogenesis of bacterial infection, and simultaneously may induce drug-resistant mutations in bacteria. As stated above, these contrasting outcomes required the urgent exploitation of specific interactions of drugs and modulation of these interactions, which can raise the susceptibility of the bacteria to therapeutic compounds [3].

Recently, marine microorganisms are important sources of novel active secondary metabolites possessing various biological activities, including antibiotics, anti-cancer, and anti-virus [4,5,6]. In the past few years, a number of biologically active compounds have been isolated from marine microorganisms, which exhibited promising pharmacological applications. Numerous marine natural products are now being tested in the clinical stages [7,8]. The recent progress of advanced technologies in genetic manufacturing and biological activity guided fractionation brought several opportunities to the discovery of marine natural products as lead compounds in the drug development [9,10].

Natural fungi are clarified as one of the huge prolific producers of secondary metabolites. The *Penicillium* genus, which comprises more than 200 species, is famous since 1920s due to the discovery of its antibiotic metabolite penicillin. This genus was discovered in some uncommon environments, including permafrost soil [11], wastewater from mines [12], and deep ocean sediments [13]. *Penicillium* species have been found to generate several bioactive compounds that could be used as antibacterial [14,15], anti-fungal [16], immunosuppressants, cholesterol-lowering agents [17], and even mycotoxins [18]. Since the discovery of penicillin, numerous *Penicillium* isolates had been examined, and new metabolites from these fungi draw a great range of attentions for scientists. These metabolites showed varied chemical structures such as polyketides [19], tetrameric acids [20], phenols and alkaloids [21] that could be potential sources for the treatment of diseases. Although the metabolites from fungi show promising biological benefits, only a limited number of the marine microbes have been investigated as bioactive agents.

Here, we focused on the extract of fermentation broth of the *Penicillium* strain M30, which was isolated from the sediment that was collected at a depth of 14 m sea at the Co To island, Northern Vietnam. This extract showed significantly antimicrobial, α-glucosidase, and α-amylase inhibitory activities. This study reports the isolation and chemical identification of secondary metabolites from the fermentation broth of the *Penicillium* strain M30 as well as their biological activity.

## 2. Results and Discussion

### 2.1. Identification of Fungus

The fungus M30 was purified successfully as described in the Materials and Methods (Figure 1). M30 strain’s colonies produce white and dry conidiophores.

It is obvious that rRNA is crucial for cell living systems. The rRNA encoding genes are extremely maintained not only in the fungal but also in other kingdoms. Moreover, proteins comprising the ribosomes and rRNA sequences are highly conserved throughout evolution. rRNA analysis is a common method used to investigate microbial diversity and to identify new strains. In this study, the M30 strain, as a member of the *Penicillium* sp., was characterized using genotyping techniques involving the amplification of the subunit of the 18S rRNA gene. A total of 1173bp of the 18S rRNA gene was sequenced and used for the identification of the isolated fungal strain. Results showed that the gene sequence of M30 strain has 99% sequence matching (Score = 2139 bits, Expect = 0.0) with 18S rRNA gene sequence of *Penicillium* (GenBank Acc. No.: MH 673731), and thus, this strain belonged to the genus *Penicillium*.

### 2.2. Biological Activities of the Extract of Penicillium sp.

In the primary experiment, M30 extract was examined for its antibacterial activity against *Escherichia coli* (ATCC 25922), *Pseudomonas aeruginosa* (ATCC 27853), *Salmonella enterica* (ATCC 12228), *Enterococcus faecalis* (ATCC 13124), *Staphylococcus aureus* (ATCC 25923), *Bacillus cereus* (ATCC 13245), and anti-yeast against *Candida albicans* (ATCC 1023). The results in Table 1 showed that M30 extract inhibited *C. albicans* with an MIC value of 64 µg/mL. This extract also exhibited moderate inhibitory activity against some Gram (+) bacteria including *S. aureus* (minimum inhibitory concentration − MIC = 256 µg/mL), *B. cereus* (MIC = 256 µg/mL) and the Gram (-) bacterium *P. aeruginosa* (MIC = 128 µg/mL). Screening results of the α-amylase and α-glucosidase inhibitory activity revealed that M30 extract exhibited a positive effect against α-glucosidase with an IC_50_ value of 628 µg/mL. In this experiment, acarbose, a positive control, showed an inhibitory effect with an IC_50_ value of 423 µg/mL. Considering the potential of the M30 strain, our subsequent study focused on the fermentation of this *Penicillium* sp. in order to analyze active secondary metabolites contributing to these above biological activities.

### 2.3. Identification of Compounds

The ethyl acetate-soluble fraction from M30 was applied to several chromatography techniques such as silica gel, reverse phase-18, and semi-preparative HPLC, resulting in the isolation of ten secondary metabolites (**1**–**10**) (Figure 2). The known compounds, sporogen AO-1 (**1**) [22], 3-indolecarbadehyde (**2**) [22], 2-[(5-methyl-1,4-dioxan-2-yl)methoxy]ethanol (**3**) [23], 2-[(2*R*-hydroxypropanoyl)amino]benzamide (**4**) [23], 4-hydroxybenzandehyde (**5**) [24], chrysogine (**6**) [24], acid 1H-indole-3-acetic (**8**) [25], and cyclo (Tyr-Trp) (**9**) [26], and 2’,3’-dihydrosorbicillin (**10**) [26] (Figure 2), were elucidated by the examination of their NMR spectra and comparison with literature data.

Compound **7** was isolated as a yellow powder. Its IR spectrum showed typical signals of hydroxyl and ketone groups at ν_max_ 3412 and 1669 cm^−1^, respectively. The high resolution electrospray ionization mass spectrometry (HR-ESI-MS) of **7** presented the pseudomolecular peak at **m/z** 189.0657 [M + H]^+^ (calculated for C_10_H_9_N_2_O_2_
**m/z** 189.0664), that could be assigned to the molecular formula of **7** is C_10_H_8_N_2_O_2_. In the ^1^H-NMR spectrum, four methine signals of a 1,2-disubstituted aromatic ring at δ_H_ 8.36 (d, *J* = 8.0 Hz, H-5), 7.63 (t, *J* = 8.0 Hz, H-6), 7.85 (t, *J* = 8,0 Hz, H-7), and 7,89 (d, *J* = 8,0 Hz, H-8), one methyl group at δ_H_ 2.77 (s, CH_3_-9), and an OH signal at δ_H_ 9.95 (brs, OH) were observed. The ^13^C-NMR and distortionless enhancement by polarization transfer (DEPT) spectrum showed that this metabolite has 10 carbon signals. Among them, a methyl signal at δ_C_ 24.0 (CH_3_-10), a ketone carbon at δ_C_ 194.1 (C-9), four methine carbon of aromatic ring, and four quaternary carbons at δ_C_ 145.3 (C-3), 160.6 (C-4), 123.5 (C-4a), and 147.7 (C-8a) were identified. In the heteronuclear multiple bond correlation (HMBC) spectrum, the long-range correlation signals between H-5 and C-8a and C-4, H-8, and C-4a were identified (Figure 3). Interestingly, the acetyl group was attached to the C-3 that was confirmed by the connection signals between CH_3_-10 and C-9 and C-3. A combination of the MS, 1D-NMR, and 2D-NMR spectrum of **7** and the comparison with the NMR data of previous literature allowed us to identify **7** as 3-acetyl-4-hydroxycinnoline. This cinnoline derivative was firstly synthesized in 1991 [27]; however, this is the first time the compound was isolated from the *Penicillium* sp.

### 2.4. Biological Activities of the Metabolites from Penicillium sp.

All of the isolated secondary metabolites (**1–10**) were tested for their antibacterial activity toward *Escherichia coli* (ATCC 25922), *Pseudomonas aeruginosa* (ATCC 27853), *Salmonella enterica* (ATCC 12228), *Enterococcus faecalis* (ATCC 13124), *Staphylococcus aureus* (ATCC 25923), *Bacillus cereus* (ATCC 13245), and anti-yeast activity against *Candida albicans* (ATCC 1023). As shown in Table 1, 2-[(5-methyl-1,4-dioxan-2-yl)methoxy]ethanol (**3**) showed strong inhibitory effect against *E. faecalis* and *C. albicans* with MIC values of 32 and 64 µg/mL, respectively. In addition, 2-[(2*R*-hydroxypropanoyl)amino]benzamide (**4**) and 4-hydroxybenzandehide (**5**) selectively suppressed *E. coli* with the MIC value of 16 µg/mL and 8 µg/mL, respectively. These compounds showed stronger inhibitory effects than the reference compound, streptomycin, that presented an MIC value of 32 μg/mL (Table 1). Previous studies revealed a strong antimicrobial property for 2-aminobenzamide derivatives and benzaldehyde derivatives [28,29,30], which have been considered as potential agents for antimicrobial drug discovery. Notably, both of 3-acetyl-4-hydroxycinnoline (**7)** and 1H-indole-3-acetic (**8**) exhibited moderate antibacterial ability against *E. faecalis* with MIC value of 64 µg/mL.

Among the tested compounds, 2’,3’-dihydrosorbicillin (**10**) potentially inhibited α-glucosidase activity at a concentration of 2.0 mM (66.31%) (Table 2). Interestingly, no α-amylase inhibitory activity was found in all the tested samples. The present study is the first study on inhibition of 2’,3’-dihydrosorbicillin (**10**) against α-glucosidase. This result suggested that 2’,3’-dihydrosorbicillin (**10**) may have potential use in the treatment of type 2 diabetes through α-glucosidase inhibition.

## 3. Materials and Methods

### 3.1. General Experiment Procedures

The ESI-MS was measured on Agilent 6120 series single quadrupole LC/MS systems (Agilent 6120, Santa Clara, CA, USA). NMR spectra were recorded on a Bruker AM500 FT-NMR spectrometer using transcranial magnetic stimulation (TMS) as an internal standard. Column chromatography (CC) was performed using a silica gel (Kiesel gel 60, 70–230 mesh and 230–400 mesh, Merck, Darmstadt, Germany). Thin layer chromatography (TLC) used pre-coated silica gel 60 F254 (Merck, Germany).

### 3.2. Fungus Isolation

The marine sediment samples were collected at 14 m depth with geographic coordinates 2105′11″–107050′57″, water at temperature 26 °C in the Co To Island, Vietnam. The samples were collected into 15 mL or 50 mL sterile Falcon tubes and preserved in an ice-box and processed within 24 h. Briefly, 0.5 g of the sample was suspended in 4.5 mL of sterile distilled water, homogenized by vortexing for 1 min, and the suspension was treated using a wet-heat technique (60 °C for 6 min). Next, 0.5 mL of this suspension was transferred to 4.5 mL sterile distilled water, and this step was repeated to set up a ten-fold dilution series to 10^−3^. At the final dilution step, aliquots of 50 µL were spread on medium A1 (soluble starch: 10 g/L; yeast extract: 4 g/L; peptone: 2 g/L; instant ocean: 30 g/L; agar: 15 g/L) supplemented with 50 µg/mL polymycin B to inhibit Gram-negative bacteria. The plates were incubated at 28 °C for seven days. The colony of fungus M30 was transferred onto a new Petri dish of medium A1 for purification.

### 3.3. Fungus Identification

Genomic DNA was extracted with the Gen Elute Bacterial Genomic DNA kit (Sigma, St. Luis, MO, USA). Sequences of 18S rRNA was used for identification of the strain. The gene amplifications were performed in a 25.0 µL mixture containing 16.3 µL of distilled water, 2.5 µL of 10X polemerase chain reaction (PCR) buffer, 1.5 µL of 25 mM MgCl_2_, 0.5 µL of 10 mM deoxynucleotide (dNTP’s), 0.2 µL of Taq polymerase, 1.0 µL for both 0.05 mM of NS3F (5′-GCAAGTCTGGTGCCAGCAGCC 3′) and 0.05 mM of NS8R (5′-TCCGCAGGTTCACCTACGGA 3′) primers and 2.0 µL of genomic DNA. The reaction tube was then put into an MJ Thermal cycler, which had been programmed to preheat at 94 °C for 3 min, followed by 30 cycles of denaturation at 94 °C for 1 min, annealing at 62 °C for 30 s, and elongation at 72 °C for 45 s before a final extension of 72 °C for 10 min. The estimated PCR product size was about 1300 bp. PCR products were purified by the DNA purification kit (Invitrogen) then sequenced by a DNA Analyzer (ABI PRISM 3100, Applied Bioscience, Carlsbad, CA, USA). Gene sequences were handled by BioEdit v.2.7.5 and compared with bacterial 18S rRNA sequences in the GeneBank database by the National Center for Biotechnology Information (NCBI) Blast program.

### 3.4. Fermentation and Extraction

Strain M30 (*Penicillium* sp.) was cultured in high-nutrient medium (30 g of instant ocean, 10 g of starch, 4 g of yeast, 2 g of peptone, 1 g of calcium carbonate, 40 mg of iron sulfate, and 100 mg of potassium bromate) for seven days at 25 °C while stirring at 200 rpm in a bioreactor (Yuin, Seoul, Korea). The culture solution (30 L) was directly extracted with EtOAc (15 L × 3 times). The EtOAc extract solution was concentrated to dryness by a rotary evaporator (Eyela, Tokyo, Japan).

### 3.5. Antimicrobial Assay

The antimicrobial assay was carried out using *E. coli* (ATCC25922), *P. aeruginosa* (ATCC27853), *S. enterica* (ATCC12228), *E. faecalis* (ATCC13124), *S. aureus* (ATCC25923), *B. cereus* (ATCC13245), and *C. albicans* (ATCC1023). Stock solutions of samples were prepared in DMSO, and the antimicrobial assays were carried out in 96-well microtiter plates against the microbial strains (5 × 10^5^ CFU/mL) using a modification of the published method. After incubation for 24 h at 37 °C, the absorbance at 650 nm was measured using a microplate reader. Streptomycin and nystatin were used as reference compounds.

### 3.6. α-Glucosidase Inhibition Assay

The inhibition of α-glucosidase activity was determined using the modified, published assay [31]. Briefly, the reaction mixture consisting of 50 μL of the sample at the indicated concentrations was incubated with 100 μL of 0.1 M potassium phosphate buffer (pH 6.8) containing α-glucosidase solution (0.5 U/mL) in 96-well plates at 37 °C for 10 min. After pre-incubation, 50 μL of 4-Nitrophenyl β-D-glucopyranoside (pNPG) was added to each well to start the reaction. After incubating at 37 °C for 10 min, absorbance readings were recorded at 405 nm in a microplate reader (Biotek, Winooski, VT, Abbr of State, USA). The control sample was added to 50 µL of buffer solution in place of the sample. Acarbose was used as a positive control of *α*-glucosidase inhibitor. The α-glucosidase inhibitory activity was calculated according to the equation below:% Inhibition = 1 − (A_sample_/A_control_) × 100(1)
where A_control_ is the absorbance of the control and A_sample_ is the absorbance of the tested samples.

### 3.7. α-Amylase Inhibition Assay

The assay was carried out following the standard protocol with slight modifications [32]. Starch azure was suspended in 0.05 M Tris–HCl buffer (pH 6.9) containing 0.01 M CaCl_2_. The tubes containing the substrate solution were boiled for 5 min and then pre-incubated at 37 °C for 5 min. A total of 100 µL of each sample and 100 µL of substrate solution and 50 µL of porcine pancreatic amylase in Tris–HCl buffer (2 units/mL) were incubated at 37 °C for 10 min. Then, 250 µL of acetic acid 50% was added in each tube to stop the reaction. After, the reaction tubes were centrifuged at 3000 rpm for 5 min at 4 °C, the absorbance of the resulting supernatant was measured at 595 nm using a microplate reader (Biotek, USA).

The α-amylase inhibitory activity was expressed as the percentage of inhibition and was calculated as follows:% Inhibition = 1 − (A_sample_/A_control_) × 100(2)
where A_control_ is the absorbance of the control and A_sample_ is the absorbance of the tested samples.

### 3.8. Isolation of Compounds

Sterilized Amberlite XAD-16 resin (2 kg) was added to fermentation broth (50 L) to absorb the extracellular secondary metabolites. The culture medium and resin were shaken for 10 h and filtered using a cheesecloth to remove the resin. The resin, cell mass, and cheesecloth were extracted with methanol overnight, concentrated under vacuum, and partitioned between water and ethyl acetate. The organic layer was dried under vacuum to afford 26.4 g of extract. The organic layer (26.4g) was separated by column chromatography (CC) eluted successively with CH_2_Cl_2_–MeOH to afford six fractions (E1–E6). Fraction E2 (5 g) was chromatographed over silica gel using *n*-hexane-acetone to get six subfractions (E2.1–E2.6). Fraction E3 (10 g) was subjected to silica gel CC and eluted with CH_2_Cl_2_–MeOH to get seven subfractions (E3.1–E3.7). Subfraction E3.1 (0.6 g) was purified by CC on Sephadex LH-20 using MeOH/CH_2_Cl_2_ (9:1, v/v) as an eluent to get sĩx subfractions (E3.1.1–E3.1.6). Fraction E3.4 (170 mg) was separated by CC on Sephadex LH-20 using MeOH to get four subfractions (E3.4.1–E3.4.4). Fraction E4 was chromatographed on an RP-18 column eluting with MeOH: water to give seven subfractions, E4.1–E4.7. Subfraction E4.2 (1.99 g) was separated over Sephadex LH-20 eluted with MeOH to get seven subfractions (E4.2.1–E4.2.7). Sub-fraction E2.2 (100 mg) was chromatographed on silica gel eluted with *n*-hexane-acetone to yield **10** (7.0 mg). Sub-fraction E3.1.2 (120 mg) was handled similar to E2.2 to yield 2.0 mg of **4** and 3.0 mg of **5**. By using CC (silica gel), sub-fraction E3.1.1, E4.2.5, and E4.2.7 was eluted individually with *n*-hexane-EtOAc, TLC CH_2_Cl_2_/EtOAc (4:6, v/v), and TLC (CH_2_Cl_2_/MeOH: 9/1) gradient to yield **2**, **8**, and **9** (5.4, 4.1 and 2.5 mg, respectively). Subfraction E3.2 (0.1 g) was purified by CC on silica gel using MeOH as an eluent to yield **3** (7.4 mg). Subfraction E3.3 (90 mg) was purified by CC on Sephadex LH-20 (MeOH), followed by preparative TLC (*n*-hexane/EtOAc/MeOH: 7/3/0.1) to furnish **6** (4 mg). Subfraction E3.4.3 (100 mg) was chromatographed on silica gel eluted with CH_2_Cl_2_–MeOH gradient, consequently gathered and checked by TLC (CH_2_Cl_2_/MeOH: 9/1), where **1** (2.3 mg) and **7** (1.6 mg) were obtained.

*Compound **1** (Sporogen AO-1)*: White amorphous powder; ^1^H-NMR (500 MHz, CDCl_3_) δ_H_ (ppm): 5.76 (d, *J* = 2,0 Hz, 1H, H-9), 5.12 (brs, 1H, H-12a), 5.10 (m, 1H, H-12b), 3.62 (td, *J* = 4,0, 10,5 Hz, H-3), 3.33 (s, 1H, H-6), 2,51 (tdd, *J* = 2.0, 5.0, 14.5 Hz, H-1a), 2.33 (dt, *J* = 2.5, 4.0, 10.5 Hz, H-1b), 2.13-2,17 (m, 1H, H-2a), 1.87 (s, 3H, H-13), 1.80-1.83 (m, 1H, H-4), 1.42-1.49 (m, 1H, H-2b), 1.26 (d, *J* = 6.5 Hz, H-15), 1.23 (s, 3H, H-14); ^13^C-NMR (125 MHz, CDCl_3_) δ_C_ (ppm): 192.8 (C-8), 162.9 (C-10), 139.1 (C-11), 121.2 (C-9), 114.5 (C-12), 71.0 (C-3), 68.3 (C-6), 64.5 (C-7), 44.4 (C-4), 41.0 (C-5), 35.2 (C-2), 31.0 (C-1), 19.8 (C-13), 18.8 (C-15), 11.3 (C-14).

*Compound **2** (3-indolecarbadehyde):* White amorphous powder; ^1^H-NMR (500 MHz, CDCl_3_) δ_H_ (ppm): 7.32 (1H, m, H-5), 7.33 (1H, m, H-6), 7.45 (1H, m, H-7), 7.85 (1H, d, *J* = 3.0 Hz, H-2), 8.32 (1H, m, H-4), 10.1 (CHO); ^13^C-NMR (125 MHz, CDCl_3_) δ_C_ (ppm): 111.4 (C-7), 120.1 (C-7a), 122.0 (C-4), 123.1 (C-5), 124.0 (C-3), 124.5 (C-6), 135.1 (C-2), 136.9 (C-3a), 185.1 (CHO).

*Compound **3** (2-((-5-methyl-1,4-dioxan-2-yl)methoxy)ethanol):* Colorless oil; IR (KBr): 3454, 2866, 1456, 1373, 1350, 1296, 1251, 1105 cm^−1^; MS-ESI: *m/z* 199.09 [M + Na]^+^ (cald. C_8_H_16_NaO_4_: **m/z** 199.09); ^1^H-NMR (500 MHz, CDCl_3_) δ_H_ (ppm): 3.59 (m, 1H, H-2), 3.63 (m, 2H, H-3), 3.62 (m, 1H, H-5), 3.35 (1H, dd, *J* = 5.0, 10.0 Hz, H-6a), 3.47 (1H, dd, *J* = 6.0, 10.0 Hz, H-6b), 3.56 (1H, m, H-1’a), 3.58 (1H, m, H-1’b), 3.59 (2H, m, H-3’), 3.59 (2H, m, H-4’), 1.10 (3H, d, *J* = 6.5 Hz, CH_3_-5); ^13^C-NMR (125 MHz, CDCl_3_) δ_C_ (ppm): 71.2 (C-2), 69.2 (C-3), 75.5 (C-5), 75.7 (C-6), 71.5 (C-1’), 71.4 (C-3’), 71.2 (C-4’), 17.7 (CH_3_-5).

*Compound **4** (2-[(2R-hydroxypropanoyl)amino]benzamide):* White amorphous powder; [α]_D_^28^ +21.2 (*c* 0.007, acetone); ESI-MS: **m/z** 231.07[M + Na]^+^; ^1^H-NMR (500 MHz, CD_3_OD) δ_H_ (ppm): 1.45 (3H, d, *J* = 6.5 Hz, H-3’), 4.25 (1H, q, *J* = 6.5 Hz, H-2’), 7.18 (1H, t, *J* = 8.0 Hz, H-5), 7.51 (1H, dt, *J* = 1.0, 8.0 Hz, H-4), 7.53 (1H, dd, *J* = 1.5, 8.0 Hz, H-6), 8.54 (1H, dd, *J* = 1.0, 8.0 Hz, H-3); ^13^C-NMR (125 MHz, CD_3_OD) δ_C_ (ppm): 21.1 (C-3’), 69.8 (C-2’), 122.2 (C-3), 122.8 (C-1), 124.5 (C-5), 129.4 (C-6), 133.3 (C-4), 139.6 (C-2), 173.3 (C-7), 176.6 (C-1’).

*Compound **5** (4-hydroxybenzaldehyde):* White amorphous powder; ^1^H-NMR (500 MHz, CDCl_3_) δ_H_ (ppm): 6.95 (2H, d, *J* = 8.5 Hz, H-aromatic), 7.80 (2H, d, *J* = 8.5 Hz, H-Phe), 9.87 (1H, s, -CHO).

*Compound **6** (Chrysogine):* Yellow powder, ^1^H-NMR (500 MHz, CDCl_3_) δ_H_ (ppm): 8.26 (1H, d, *J* = 8.0 Hz, H-3), 7.76 (1H, t, *J* = 7.5 Hz, H-5), 7.67 (1H, d, *J* = 8.0 Hz, H-6), 7.47 (1H, t, *J* = 7.5 Hz, H-4), 4.87 (1H, q, *J* = 6.5 Hz, H-9), 1.65 (3H, d, *J* = 7.0 Hz, H-10). ^13^C-NMR (125 MHz, CDCl_3_) δ_C_ (ppm): 162.8 (C-1), 157.9 (C-8), 148.6 (C-7), 134.9 (C-4), 127.2 (C-6), 126.9 (C-5), 126.5 (C-3), 121.0 (C-2), 67.5 (C-9), 22.5 (C-10).

*Compound **7** (3-Acetyl-4-hydroxycinnoline):* Yellow powder; IR ν_max_ 3412 and 1669 cm^−1^; HR-ESI-MS **m/z** 189.0657 [M+H]^+^ (calculated for C_10_H_9_N_2_O_2_
**m/z** 189.0664); ^1^H-NMR (500 MHz, CDCl_3_) δ_H_ (ppm): 9.95 (1H, brs, OH), 8.36 (1H, d, *J* = 8.0 Hz, H-5), 7.89 (1H, d, *J* = 8.0 Hz, H-8), 7.85 (1H, t, *J* = 8.0 Hz, H-7), 7.63 (1H, t, *J* = 8.0 Hz, H-6), 2.77 (3H, s, H-10); ^13^C-NMR (125 MHz, CDCl_3_) δ_C_ (ppm): 24.0 (C-10), 123.5 (C-4a), 126.9 (C-5), 129.2 (C-8), 129.4 (C-6), 134.9 (C-7), 145.3 (C-3), 147.7 (C-8a), 160.6 (C-4), 194.1 (C-9).

*Compound **8** (1H-Indole-3-acetic acid):* White amophous powder; ^1^H-NMR (500 MHz, CD_3_OD) δ_H_ (ppm): 3.62 (1H, s, CH_2_), 6.98 (1H, dt, *J* = 1.0, 8.0 Hz, H-5), 7.06 (1H, dt, *J* = 1.0, 8.0 Hz, H-6), 7.15 (1H, s, H-2), 7.31 (1H, dd, *J* = 0.5, 8.0 Hz, H-7), 7.62 (1H, d, *J* = 0.5, 8.0 Hz, H-4).

*Compound **9** (Cyclo (Tyr-Trp)):* White amophous powder; ^1^H-NMR (500 MHz, MeOD) δ_H_ (ppm): 1.48 [1H, dd, *J* = 8.5, 13.5 Hz, CH_2a_(Tyr)], 2.57 [1H, dd, *J* = 4.0, 14.0 Hz, CH_2b_(Tyr)], 2.76 [1H, dd, *J* = 6.0, 14.5 Hz, CH_2a_(Trp)], 3.05 [1H, dd, *J* = 4.0, 14.5 Hz, CH_2b_(Trp)], 3.87 [1H, dd, *J* = 3.5, 8.5 Hz, H-3), 4.16 (1H, t, *J* = 5.0 Hz, H-6), 6.46 (2H, d, *J* = 7.5 Hz, H-Tyr), 6.63 (2H, d, *J* = 7.5 Hz, H-Tyr), 7.03-7.60 (5H, H-Trp); ^13^C-NMR (125 MHz, MeOD) δ_C_ (ppm): 31.2 (CH_2_), 40.6 (CH_2_), 57.1 (CH), 57.9 (CH), 109.6-157.5 (C-Tyr và C-Trp), 169.3 (C=O), 169.7 (C=O).

*Compound **10** (2’,3’-dihydrosorbicillin):* Yellow powder; ^1^H-NMR (500 MHz, CD_3_OD) δ_H_ (ppm): 7.59 (s, 1H, H-6), 5.53-5.54 (m, 2H, H-4’ and H-5’), 3.06 (t, *J* = 7.5 Hz, H-2’), 2.35-2.43 (m, 2H, H-3’), 2.39 (s, 3H, H-7), 2.27 (s, 3H, H-8), 1.65 (m, 3H, H-6’); ^13^C-NMR (125 MHz, CD_3_OD) δ_C_ (ppm): 207.4 (C-1’), 161.5 (C-2), 156.7 (C-4), 130.9 (C-6), 130.1 (C-4’), 127.0 (C-5’), 124.9 (C-5), 122.8 (C-3), 117.3 (C-1), 39.2 (C-2’), 28.6 (C-3’), 18.0 (C-6’), 17.2 (C-7), 10.3 (C-8).

### 3.9. Statistical Analysis

Data were expressed as the mean ± standard deviations (SD). Statistical significance was assessed by the two-tailed unpaired Student’*s t*-test, and P values less than 0.05 were considered statistically significant.

## 4. Conclusions

In recent years, ordinary natural-derived *Penicillium* sp. is among the most attentive fungi in the science of natural products. In our experiments, we identified the *Penicillium* strain M30 from sediment collecting at a depth of 14 m at the Co To island, the North of Vietnam’s sea. After fermentation, we found that the extract of fermentation broth of this sp. showed significant antimicrobial and α-glucosidase/α-amylase inhibitory activities. Extraction and isolation results yielded a new natural product, 3-acetyl-4-hydroxycinnoline (**7**), and nine known metabolites as sporogen AO-1 (**1**), 3-indolecarbadehyde (**2**), 2-[(5-methyl-1,4-dioxan-2-yl)methoxy]ethanol (**3**), 2-[(2*R*-hydroxypropanoyl)amino]benzamide (**4**), 4-hydroxybenzandehyde (**5**), chrysogine (**6**), acid 1H-Indole-3-acetic (**8**), cyclo (Tyr-Trp) (**9**), and 2’,3’-dihydrosorbicillin (**10**). Among the isolates, 2-[(5-methyl-1,4-dioxan-2-yl) methoxy]ethanol (**3**) showed a strong inhibitory effect against *E. faecalis* with a MIC value of 32 µg/mL. Both of 2-[(2*R*-hydroxy propanoyl) amino] benzamide (**4**) and 4-hydroxybenzandehyde (**5**) selectively killed *E. coli* with MIC values of 16 and 8 µg/mL, respectively. 2’, 3’-Dihydrosorbicillin (**10**) potentially inhibited activity at a concentration of 2.0 mM (66.31% α-glucosidase). This study highlights the value of *Penicillium* sp. derivation from sediment as a source of antimicrobial, antifungal, and anti-α-glucosidase compounds with the potential to combat afflicting pathogens and diabetic diseases.

## Figures and Tables

**Figure 1 molecules-24-03830-f001:**
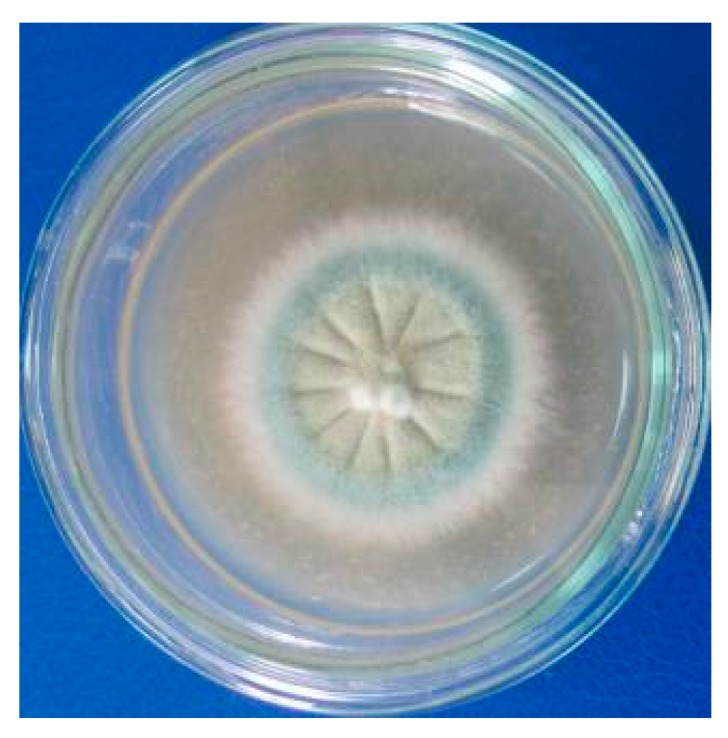
Appearance of M30 strain’s colonies.

**Figure 2 molecules-24-03830-f002:**
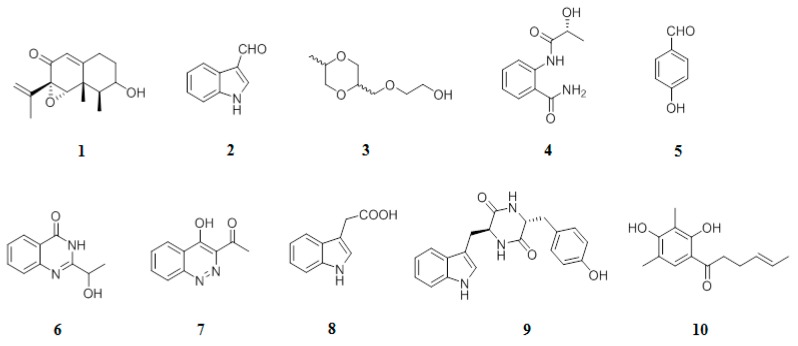
Chemical structures of compounds **1–10**.

**Figure 3 molecules-24-03830-f003:**
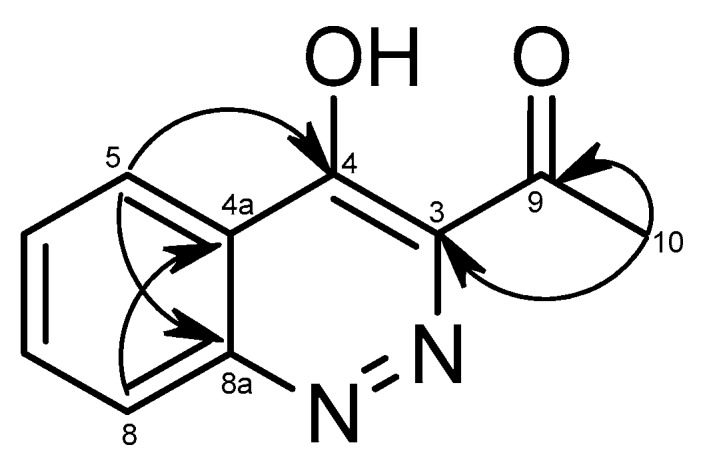
Selected key correlations in the HMBC spectrum (
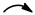
) of 7.

**Table 1 molecules-24-03830-t001:** Minimum inhibitory concentration values of extract and isolated metabolites from *Penicillium* sp.

Samples/Compounds	Gram (+) ^(a)^			Gram (-) ^(a)^			Yeast ^(a)^
	*E. faecalis* ATCC29212	*S. aureus* ATCC25923	*B. cereus* ATCC13245	*E. coli* ATCC25922	*P.aeruginosa* ATCC27853	*S. enterica* ATCC13076	*C. albicans* ATCC10231
**M30 extract**	-	256	256	-	128	-	64
**1**	256	>256	>256	>256	>256	>256	128
**2**	256	>256	>256	>256	>256	>256	128
**3**	32	>256	>256	>256	>256	>256	64
**4**	>256	>256	>256	16	128	>256	>256
**5**	>256	>256	>256	8	>256	>256	>256
**6**	>256	>256	>256	64	>256	>256	>256
**7**	64	256	256	>256	>256	>256	128
**8**	64	128	256	>256	128	256	128
**9**	256	>256	>256	>256	>256	>256	>256
**10**	>256	>256	>256	>256	>256	>256	>256
**Streptomycin ^(b)^**	256	256	128	32	256	128	-
**Cyclohexamide ^(b)^**	-	-	-	-	-	-	32

^(a)^ Results are calculated in µg/mL. ^(b)^ Positive compounds.

**Table 2 molecules-24-03830-t002:** α-Glucosidase inhibitory activity of compounds **1**–**10**.

Sample	Concentration (mM)	% Inhibition^(a)^
**M30**	100 µg/mL	-
	500 µg/mL	44.78 ± 1.73
**1**	0.5	-
	2.0	-
**2**	0.5	-
	2.0	-
**3**	0.5	-
	2.0	-
**4**	0.5	12.19 ± 2.93
	2.0	18.44 ± 3.94
**5**	0.5	14.45 ± 1.28
	2.0	18.80 ± 0.54
**6**	0.5	-
	2.0	-
**7**	0.5	9.42 ± 1.03
	2.0	18.81 ± 1.97
**8**	0.5	19.79 ± 0.83
	2.0	40.31 ± 0.50
**9**	0.5	17.91 ± 1.13
	2.0	39.23 ± 0.68
**10**	0.5	12.11 ± 0.60
	2.0	66.31 ± 0.09
Acarbose ^(b)^	100 µg/mL	19.69 ± 2.28
	500 µg/mL	55.18 ± 0.67

^(a)^ The experiments were carried out in three replicates. ^(b)^ Positive compound.

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
