# Peer review of "Chemical Composition and Biological Activities of Metabolites from the Marine Fungi Penicillium sp. Isolated from Sediments of Co To Island, Vietnam"

_molecules, 2019, doi:10.3390/molecules24213830_

Round 1

Reviewer 1 Report

The manuscript focus on the characterization and biological evaluation of the extract of fermentation broth of Penicillium strain M30.

In general the manuscript is fluent, the topic in interesting and it can be published in the journal. It is, in fact well known the value of marine environments as relevant source of biologically active molecules.

Some minor points:

Introduction is relatively weak and will benefit from a discussion on the global threat of antibiotic-resistance and on the urgent need to find new therapeutic strategies to overcome this problem. Add the recent references on the topic: Science, 2016, 351, 6268, aad3292, European Journal of Medicinal Chemistry, 2019, 161, 154-178. Bibliography on the important role of marine microorganisms as valuable sources of biologically active molecules needs an update: add recent ref. European Journal of Medicinal Chemistry, 2017, 138, pp. 371-383 in line 35 after “anticancer” and Curr Med Chem., 2016;23(25):2892-2905 after “antibiotics”. Spectroscopic data must be checked, there are some signals missing Conclusion: author should deepen this section that appears weak

Author Response

Manuscript ID: molecules-614604

Type: Article

Number of Pages: 11

Title: Chemical composition and biological activities of marine fungi Penicillium sp. isolated from sediments of Co To island, Vietnam

Authors

Hong Minh Thi Le * , Quynh Thi Do , Mai Huong Thi Doan , Quyen Thi Vu , Mai Anh Nguyen , Thu Huyen Thi Vu , Hai Dang Nguyen , Nguyen Thi Thuy Duong , Manh Hung Tran * , Van Minh Chau , Van Cuong Pham

Reviewer 1:

Introduction is relatively weak and will benefit from a discussion on the global threat of antibiotic-resistance and on the urgent need to find new therapeutic strategies to overcome this problem. Add the recent references on the topic: Science, 2016, 351, 6268, aad3292, European Journal of Medicinal Chemistry, 2019, 161, 154-178. Bibliography on the important role of marine microorganisms as valuable sources of biologically active molecules needs an update: add recent ref. European Journal of Medicinal Chemistry, 2017, 138, pp. 371-383 in line 35 after “anticancer” and Curr Med Chem., 2016;23(25):2892-2905 after “antibiotics”. Spectroscopic data must be checked, there are some signals missing Conclusion: author should deepen this section that appears weak

Answer: Thank you for your kind recommendation. We had already added that information into our introduction part. Please see our new revision text on Page 1-2, line 34-47.

Reviewer 2 Report

This work describes the characterization and biological activities of metabolites from Penicillium sp.

The overall scientific content is acceptable, but not in the current format. Extensive English changes are required, especially in Introduction and Results and discussion.

In my opinion the paper can be accepted after some modifications:

1. References should be checked:

- The name of the microorganism needs to be written in italic (ref. 7, 10, 15, 16),

- References should include DOI numbers.

- Ref. 1: Space should be deleted.

- Ref. 9 and 10: Quotation marks should be deleted in the title.

- Ref. 24: 13 should be deleted.

2. Title should be modified: Chemical composition and biological activities of metabolites from the marine fungi Penicillium sp. isolated from sediments of Co To island, Vietnam.

3. Graphical abstract should be included.

4. Section Materials and Methods. Isolation of compounds:

- The first paragraph on page 4 should be simplified.

- Pag. 4, lines 160, 177, 184: it is written “500 MHz” instead of “125 MHz”.

- Pag. 5, lines 190, 204: it is written “500 MHz” instead of “125 MHz”.

5. Results and discussion:

- Pag. 7, line 266: It is written “Conbination” instead of “Combination”. Number 7 should be written in bold.

- Pag. 7, line 268: Reference 20 does not correspond to the text.

- Tables 1 and 2 should be modified: first, one table referring only to the extract should be included, so that the explanation is clearer.

6. Conclusion: The font size of “Penicillium” is larger.

7. Supplementary material with information about NMR data and biological assays should be included.

In summary, this paper includes useful information but requires some modifications to make it suitable for publication.

Author Response

Manuscript ID: molecules-614604

Type: Article

Number of Pages: 11

Title: Chemical composition and biological activities of marine fungi Penicillium sp. isolated from sediments of Co To island, Vietnam

Authors: Hong Minh Thi Le * , Quynh Thi Do , Mai Huong Thi Doan , Quyen Thi Vu , Mai Anh Nguyen , Thu Huyen Thi Vu , Hai Dang Nguyen , Nguyen Thi Thuy Duong , Manh Hung Tran * , Van Minh Chau , Van Cuong Pham

Reviewer 2

References should be checked:

 - The name of the microorganism needs to be written in italic (ref. 7, 10, 15, 16),

- References should include DOI numbers.

- Ref. 1: Space should be deleted.

- Ref. 9 and 10: Quotation marks should be deleted in the title.

- Ref. 24: 13 should be deleted.

 Answer 1:

Scientific names of species have already corrected (Penicillium mycobiota, Penicillium ochrochloron, Penicillium, Penicillium). Please see somewhere in the main text. Space of reference 1 was deleted. Quotation marks were deleted in the titles. Ref 24 had been formatted.

Title should be modified: Chemical composition and biological activities of metabolites from the marine fungi Penicilliumsp. isolated from sediments of Co To island, Vietnam.

Answer 2: Title modification has been done. Please see page 1.

Graphical abstract should be included.

Answer 3: Graphical abstract is not required so that we would like to ommit this one.

Section Materials and Methods. Isolation of compounds:

- The first paragraph on page 4 should be simplified.

- Pag. 4, lines 160, 177, 184: it is written “500 MHz” instead of “125 MHz”.

- Pag. 5, lines 190, 204: it is written “500 MHz” instead of “125 MHz”.

 Answer 4: Isolation part was revised. Please see page 4, line 144-166.

The others were revised. Please see the appropriate position with yellow highlighted text.

Results and discussion:

 - Pag. 8, line 278: It is written “Conbination” instead of “Combination”. Number 7 should be written in bold.

- Pag. 7, line 268: Reference 20 does not correspond to the text.

- Tables 1 and 2 should be modified: first, one table referring only to the extract should be included, so that the explanation is clearer.

 Answer 5:

Spelling of conbination had been corrected in line 278. Number 7 in line 278 is in bold. Reference 20 was moved, and exchanged by reference 29. Please see page 11, line 397. We would like to keep Tables 1 and 2 as the presence form. They are easy to see and understandable.

Conclusion: The font size of “Penicillium” is larger.

 Answer 6: Thanks. Please see page 10, line 308.

Supplementary material with information about NMR data and biological assays should be included.

 Answer 7: NMR data is listed in the main text. Please see pages 4-5.